# Effects of Botulinum Toxin Type A on the Nociceptive and Lemniscal Somatosensory Systems in Chronic Migraine: An Electrophysiological Study

**DOI:** 10.3390/toxins15010076

**Published:** 2023-01-14

**Authors:** Gabriele Sebastianelli, Francesco Casillo, Antonio Di Renzo, Chiara Abagnale, Ettore Cioffi, Vincenzo Parisi, Cherubino Di Lorenzo, Mariano Serrao, Francesco Pierelli, Jean Schoenen, Gianluca Coppola

**Affiliations:** 1Department of Medico-Surgical Sciences and Biotechnologies, Sapienza University of Rome Polo Pontino-ICOT, 04100 Latina, Italy; 2IRCCS-Fondazione Bietti, 00198 Rome, Italy; 3Headache Research Unit, CHU de Liège, Neurology, Citadelle Hospital, B-4000 Liège, Belgium

**Keywords:** botulinum toxin type A, migraine, peripheral sensitization, central sensitization, trigemino-cervical complex, pain, lemniscal system

## Abstract

(1) Background: OnabotulinumtoxinA (BoNT-A) is a commonly used prophylactic treatment for chronic migraine (CM). Although randomized placebo studies have shown its clinical efficacy, the mechanisms by which it exerts its therapeutic effect are still incompletely understood and debated. (2) Methods: We studied in 15 CM patients the cephalic and extracephalic nociceptive and lemniscal sensory systems using electrophysiological techniques before and 1 and 3 months after one session of pericranial BoNT-A injections according to the PREEMPT protocol. We recorded the nociceptive blink reflex (nBR), the trigemino-cervical reflex (nTCR), the pain-related cortical evoked potential (PREP), and the upper limb somatosensory evoked potential (SSEP). (3) Results: Three months after a single session of prophylactic therapy with BoNT-A in CM patients, we found (a) an increase in the homolateral and contralateral nBR AUC, (b) an enhancement of the contralateral nBR AUC habituation slope and the nTCR habituation slope, (c) a decrease in PREP N-P 1st and 2nd amplitude block, and (d) no effect on SSEPs. (4) Conclusions: Our study provides electrophysiological evidence for the ability of a single session of BoNT-A injections to exert a neuromodulatory effect at the level of trigeminal system through a reduction in input from meningeal and other trigeminovascular nociceptors. Moreover, by reducing activity in cortical pain processing areas, BoNT-A restores normal functioning of the descending pain modulation systems.

## 1. Introduction

Chronic migraine (CM) is a severely disabling form of headache with a drastic impact on quality of life [1,2] and with a high socioeconomic burden [3,4,5,6]. The prevalence of chronic migraine is generally assessed to be 8% in migraine patients and around 1–2% in the general population [3,7,8]. Epidemiologic studies over the course of 1 year have shown a progression rate of 3% among patients with episodic migraine [9,10]. Most of these patients also have medication overuse headache (MOH) [10], which is considered together with obesity, depression, inefficacy of acute treatment, and stressful life events the most important factors of conversion from episodic to CM [11]. The disability and health effects associated with chronic episodes of headaches highlight the value of preventive pharmacotherapy, which aims to reduce the frequency, severity, duration, and disability of chronic migraine. OnabotulinumtoxinA (BoNT-A) has been shown to alleviate pain in a number of conditions, including migraine [12]. The PREEMPT (Phase III REsearch Evaluating Migraine Prophylaxis Therapy) clinical trial evaluated the safety and efficacy of BoNT-A in adult migraine patients and found that, compared with placebo, it reduced the mean frequency of headache days [13].

Although randomized placebo studies have clearly shown the clinical efficacy of BoNT-A, knowledge about the mechanisms by which it is able to prevent chronic migraine is still incomplete and debated. Apart from BoNT-A’s well-known ability to inhibit the release of acetylcholine at the neuromuscular junction, there is much evidence of its action in reducing the release of pro-inflammatory substances and neurotransmitters, thus preventing peripheral sensitization at the level of the first-order trigeminovascular system [14,15,16,17,18,19,20,21,22,23], and blocking the release of neuropeptides, such as calcitonin gene-related peptide (CGRP), at the level of second-order trigeminal neurons in the brainstem and trigeminal ganglia [24,25]. Whether the suppression of peripheral sensitization triggering mechanisms by BoNTA administration is also secondarily able to inhibit the development and maintenance of central sensitization, one of the mechanisms underlying the chronification of migraine, is still unknown. Moreover, it is not yet known if this neuromodulatory effect is also transmitted at the level of cortical nociceptive receptive fields and in non-painful somatosensory systems. For these purposes, we studied, in a group of chronic drug-resistant migraine patients, the trigeminal and extratrigeminal system at different levels through the use of electrophysiological techniques before and 1 and 3 months after a single injection session with BoNTA, according to the PREEMPT protocol [26]. Specifically, we electrically stimulated the supraorbital region and simultaneously recorded the nociceptive blink reflex, which reflects the activity of the caudal trigeminal nucleus [27]; the trigemino-cervical reflex, which studies the integrity of cervical motoneurones related to the trigeminal system [28]; and the pain-related cortical evoked potential, which reveals the activity of the cingulate area [29]. In addition, for each patient, we acquired somatosensory cortical evoked potentials from median nerve stimulation at the wrist, a way to assess the integrity of the non-pain-related somatosensory lemniscal system [27]. Based on the results obtained in the animal model and in humans, we hypothesize that a single injection of BoNTA may induce a desensitization of the trigeminal nociceptive system at the peripheral level and a normalization of responses after repeated stimulation. Furthermore, we argue that these effects are confined to the cephalic and not extracephalic level.

## 2. Results

### 2.1. Clinical Data

Administration of BoNT-A significantly reduced the number of headache days at T1 (between-group test *p* = 0.014) and T3 (between-group test *p* < 0.001) compared to T0. In addition, BoNT-A also significantly decreased acute medication intake at T1 (between-group test *p* = 0.006) and T3 (between-group test *p* < 0.001) compared to T0. Headache severity significantly diminished 3 months (T3 vs. T0: *p* = 0.001), but not 1 month (T1 vs. T0: *p* = 0.301), after BoNT-A (Table 1).

### 2.2. Nociception Specific Blink Reflex (nBR)

ANOVA statistics revealed a significant repeated time measurements effect for the mean pain threshold after supraorbital stimulation (F = 3.23; *p* < 0.05), which significantly increased at T3 compared to T0 (between-group test *p* = 0.04), while the mean sensory detection threshold did not change significantly (S = 1.02, *p* = 0.599) (Table 2).

Statistics revealed a significant repeated time measurements effect for the AUC of the 1st ipsilateral (S = 6.00, *p* = 0.05) and contralateral nBR block (F = 5.31, *p* = 0.01). Between-group tests revealed that the AUC was significantly increased at T3 compared to T0 for the ipsilateral 1st block (Dunn–Bonferroni test = 5.73, *p* = 0.05) and for all the three contralateral blocks (1st block *p* = 0.010; 2nd block *p* = 0.040; 3rd block *p* = 0.042) (Figure 1).

A significant repeated time measurements effect was found for the habituation slope calculated between the 1st and the 2nd block of the contralateral (S = 6.03, *p* = 0.049), but not of the ipsilateral nBR AUC (S = 1.98, *p* = 0.151). The between-group analysis revealed that the contralateral habituation slope at the 2nd block was significantly more pronounced at T3 than at T0 (*p* = 0.049). The repeated time measurements effect was not significant for the nBR AUC habituation slope calculated between the 1st and the 3rd nBR blocks, both for ipsilateral (S = 1.28, *p* = 0.528) and contralateral (F = 1.19, *p* = 0.316) responses (Table 2, Figure 1).

### 2.3. Nociception Specific Trigemino-Cervical Reflex (nTCR)

There was no significant repeated time measurement effect for nTCR onset latency (F = 0.41, *p* = 0.664), duration (F = 1.23, *p* = 0.303), grand-average AUC (S = 2.60, *p* = 0.273), 1st block AUC (S = 2.22, *p* = 0.330), or habituation slope calculated between the 1st and 2nd block (S = 1.38, *p* = 0.5). Only the nTCR habituation slope between the 1st and 3rd block (S = 5.71, *p* = 0.028) showed a repeated time measurements effect (Table 3). Between-group analysis revealed that the nTCR habituation slope was more pronounced at T3 than at T0 (*p* = 0.024) (Table 3, Figure 1).

### 2.4. Pain-Related Evoked Potentials (PREP)

A significant repeated time measurements effect was observed for PREP N-P 1st and 2nd block amplitude (F = 3.40, *p* = 0.043; F = 4.77, *p* = 0.014, respectively). On the between-group test, the N-P 1st and 2nd block amplitudes were significantly reduced (*p* < 0.05, *p* = 0.025, respectively) at T3 compared to T0.

We found no significant repeated time measurements effect for N (S = 2.85, *p* = 0.240) and P (F = 0.22, *p* = 0.802) latencies, or PREP amplitude habituation at 2nd (F = 2.87, *p* = 0.069) and 3rd (F = 0.30, *p* = 0.746) block of averaging (Table 4, Figure 2). 

### 2.5. Somatosensory Evoked Potentials (SSEP)

None of the latency and amplitude parameters of the various SSEP components showed a significant repeated time measurements effect, with the same being true for the habituation slope measured at 2nd and 3rd blocks (see Table 5, Figure 2).

### 2.6. Correlation Analyses

The correlation analysis showed no statistically significant relationship between the percentage changes in neurophysiological variables at 1 or 3 months and the percentage changes in clinical variables (days with headache/month, mean pain severity and number of acute medications/month).

## 3. Discussion

Our study provides electrophysiological evidence for the ability of a single session of BoNT-A injections with the PREEMPT protocol in CM patients to exert a neuromodulatory effect of the nociceptive trigeminal system at the level of the brain stem and the cerebral cortex. Three months after the BoNT-A injections, the most significant findings were: (a) an increase in the supraorbital pain threshold (PT), but not the sensory detection threshold (ST); (b) an increase in ipsilateral (1st block) and contralateral (1st, 2nd, and 3rd block) nBR AUC; (c) increased habituation of the contralateral nBR AUC (between the 1st and 2nd block) and of nTCR (between the 1st and 3rd block); (d) a decrease in PREP N-P 1st and 2nd block amplitude. By contrast, we found no effect of BoNT-A on the non-noxious somatosensory system after extracephalic stimulation.

BoNT-A is a 900 kDa protein complex with an active portion composed of a heavy chain (100 kDa) involved in membrane translocation, linked by a disulfide bridge to a light chain (50 kDa) responsible for the catalytic intracellular activity [30]. After internalization into the cell and the synaptic vesicles, the light chain cleaves a synaptosomal-associated protein (SNAP-25), a crucial component of the SNARE complexes, hence preventing the fusion of synaptic vesicles to the inner surface of the cell membrane [30]. The best-known action of botulinum toxin is on peripheral motor nerves where it inhibits acetylcholine (ACh) release at neuromuscular junctions provoking muscle paralysis. However, the capacity of BoNT-A to reduce muscle pain does not always correlate with its ability to cause muscle paralysis [14]. Interfering with the synaptic vesicle cycle, BoNT-A also inhibits the release of other neurotransmitters, such as glutamate [15] and neuropeptides (e.g., CGRP [16], substance P [17,18], PACAP-38), and the insertion of receptors and ion channels (eg, TRPA1, TRPV1 [19], P2X3 [20]) into neuronal membranes [21]. Some of these neuropeptides and receptors/ion channels are involved both in pain perception and in migraine pathophysiology. Focusing on CGRP, in vitro animal experiments showed that BoNTA inhibits the release of CGRP from sensory neurons [16,31] while in a clinical study treatment with BoNT-A decreased interictal CGRP plasma levels in CM patients who were treatment responders [32]. It has been shown that in primary sensory neuron cultures, the toxin is able to block KCl-evoked release of substance P and CGRP [16,17] and when intradermally injected in humans it reduces capsaicin- and heat-evoked glutamate release [33]. BoNT-A also decreases the insertion of pain-related ion channels such as TPRV1 or TRPA1 in the membrane of first-order sensory neurons, the upregulation of which may be responsible for the reduced pain threshold (sensitization) that is associated with migraine attacks [34]. Moreover, a recent study by Gfrerer et al. suggests that BoNT-A may also act as a migraine preventive therapy by reducing the inflammation via the modulation of inflammatory gene expression and immune cells [35]. In conclusion, the therapeutic effectiveness of BoNT-A in migraine prevention may be explained by the simultaneous lowering of CGRP release, sensitivity to molecules that activate nociceptive meningeal C-afferents via TRPV1 and TRPA1, and pre-existing inflammation. It is worth noting that BoNT-A inhibits mechanical nociception to suprathreshold stimuli in peripheral trigeminal neurons acting on C but not on Aδ meningeal nociceptors [36]. These effects on unmyelinated C-fibers have been reported for extracranial and intracranial meningeal nociceptors [36]. The identification of distinct populations of sensory fibers that pass between the outer and inner portions of the calvarial bones through calvarian sutures provides a significant explanation for how extracranial injections of BoNT-A are able to prevent activation of intracranial meningeal nociceptors. Two main fiber pathways were discovered: one belonging to the trigeminovascular system, originating in the trigeminal ganglion [37], and a second one belonging to the cervicovascular pathway, originating in cervical C2-C3 dorsal root ganglia [38]. 

Here, we studied on purpose the nociceptive blink reflex with a concentric surface electrode able to depolarize the superficial layer of the dermis and chiefly activate Aδ fibers [28,39]. Therefore, it seems likely that the observed nBR changes are not due to an effect of BoNTA of the peripheral nerve fibers where it targets C afferents [36], but merely on the trigeminal nucleus.

Few studies have analyzed the nBR in CM patients in baseline conditions and their results are not concordant. De Marinis et al. found no difference between CM patients without medication overuse, recorded during and outside of an attack, and control subjects, whereas they found a lack of the paired-pulse recovery cycle for the R2 AUC of the conventional BR, i.e., a more pronounced inhibition at the level of the spinal trigeminal nucleus (STN), especially in patients recorded outside an attack [40]. Contrasting with these results, Sohn et al. showed that compared to healthy volunteers and episodic migraine patients between attacks, CM patients have smaller baseline amplitudes and AUC values for the nBR and larger amplitude of PREP compared to the controls [41]. They also found a negative correlation between amplitude and AUC of the nBR and monthly number of headache days. 

If we assume that our MOH+CM patients had baseline electrophysiological characteristics in line with previous studies, namely a decreased amplitude of initial nBR blocks and an increased amplitude of PREPs, the increase in ipsi- and contralateral nBR AUC and the decrease in PREP N-P 1st and 2nd block amplitudes we have found 3 months after the BoNT-A injections could represent the electrophysiological counterpart of a normalized function of the entire trigeminovascular system induced by BoNT-A. That BoNT-A is able to revert in CM the sensitization of the 1st order trigeminovascular neurons [36,42] by reducing the firing of peripheral C-fibers and inhibiting the release of CGRP [16,17] is also suggested by the increase in PT after three months. 

In addition, we found that habituation of nBR and nTCR that was deficient at baseline, normalizes after BoNT-A. This suggests that, by decreasing the afferent firing, BoNT-A could enhance the inhibitory antinociceptive activity of the brainstem-STN on sustained nociceptive stimulations, which in turn would be responsible for the progressive reduction in response magnitude of the trigeminocervical system [27]. The fact that the BoNT-A injections normalize contralateral, but not ipsilateral, nBR habituation could be related to the previous observation of an inverse relationship between initial (1st block) amplitude and the degree of late habituation [43,44]. The rise of AUC of the 1st nBR contralateral block at T3 as compared to that of T0 (T3–T0 = 0.278) is indeed greater than that of the ipsilateral one (T3–T0 = 0.168) and thus more prone to habituation (see Table 2). 

It is well known that the brainstem synaptic transmission and the excitability of brainstem interneurons [45], which are known to be under the effect of suprasegmental regulation, mostly from the cerebral cortex and basal ganglia [46,47], are reflected in the R2 component of the blink reflex. It has been suggested that the R2 amplitude decrease at the brainstem level in CM patient could be related to impaired descending pain modulation, which occurs during migraine chronification [41]. Central sensitization and defective central pain control systems are thought to promote the development of chronic pain [48]. We hypothesize that BoNT-A, by reverting peripheral sensitization and reducing the number of headache/months, could reduce the facilitation of trigeminal pain processing at the cortical level [41], as documented by the reduction in PREP 1st block amplitudes and further amplitude reduction during stimulus repetition, i.e., habituation. This might normalize the descending pain modulation pathways leading to reduced suppression of brainstem interneurons and hence increase in ipsi- and contralateral nBR AUC. 

It is important to note that these effects are limited to the nociceptive cephalic system since we see no effect of the BoNT-A treatment on somatosensory potentials evoked by non-noxious median nerve stimulation. Partially in line with our results, de Tommaso et al. [49] reported that 1 week after a single injection session of BoNT-A, the baseline PREP habituation deficit normalized when the potential was elicited by laser stimulation to the supraorbital region, but not by stimulation of the hand. Such a rapid electrophysiological effect is likely due to the ability of the laser stimulation to exclusively activate nociceptive C fibers, as opposed to our galvanic stimulation which activates nociceptive A-delta and non-nociceptive A-beta fibers.

Contrary to its well-known peripheral mechanism of action, the activity of BoNT-A on the central nervous system (CNS) is still debated. There is evidence suggesting that central antinociceptive effects of BoNT-A could be mediated by an increase in opioidergic [50] and GABAergic [51] neurotransmission, which can occur by axonal transport of the toxin via sensory afferents to nociceptive nuclei in the CNS [52,53,54]. These data, however, are contradicted by recent studies denying the possibility of a transsynaptic transfer of BoNT-A [55,56], and supporting the hypothesis that central desensitization, synaptic plasticity, and other CNS effects of BoNT-A are secondary phenomena due to the decreased peripheral inputs rather than to a direct central effect [57]. In animal models, extracranial injection of BoNT-A cannot prevent cortical spreading depression, an electrocortical phenomenon thought to mediate the migraine aura, but can attenuate the overall firing of unmyelinated C meningeal fibers [38]. In our study, the absent correlation between trigeminal (nBR and nTCR) and central (PREP) nociceptive electrophysiological responses and clinical changes suggests that the clinical efficacy of BoNT-A is mainly due to peripheral modulation of the trigeminovascular sensory system. Taken together, these findings suggest that BoNT-A works by reverting peripheral sensitization, resulting in a decrease in peripheral firing and a modulatory effect on the entire trigeminocervical system and cortical nociceptive areas.

We acknowledge that our study has some limitations. First, we have not recorded our patients at later time points than 3 months after the injection. We cannot exclude therefore that a wearing-off of the BoNT-A effect, that has been shown to start losing its action during the 3rd month after injection, might have influenced our neurophysiological results [58]. However, the fact that the pain threshold increase and the PREP amplitude decrease are still there after 3 months disproves this. Another potential weakness of the study is that we did not use a laser stimulation, which would have been useful for analyzing the effect of BoNT-A on C-fibers and corroborate our hypothesis. Finally, the relatively small sample size due to the interrupted availability of BoNT-A in our hospital is another limitation. 

Further studies with a greater sample size and with extension of recording sessions up to 6 months after the first BoNT-A dose could be performed to support our results.

## 4. Conclusions

In conclusion, we show that a single session of BoNT-A injections in chronic migraine patients according to the PREEMPT protocol, besides clinical improvement, after 3 months results in recordable excitability changes of the trigeminal nociceptive system at the level of the brain stem and the cortex. The observed changes can be explained by a reduction in input from meningeal and other trigeminovascular nociceptors, which normalizes the effect of descending pain modulation systems on the brain stem and reduces activity of cortical pain processing areas.

## 5. Materials and Methods

### 5.1. Subjects

Fifteen patients with medication overuse headache (MOH) and chronic migraine, who received diagnosis in accordance with the diagnostic criteria of the International Classification of Headache Disorders (ICHD third edition) [59], were recruited among consecutive patients attending the Headache clinic of Sapienza University of Rome Polo Pontino. We collected information about the patients’ clinical characteristics: age at headache onset, years of disease, monthly attack frequency (n/month), attack duration (hours), monthly number of tablets intake (n/months), and type and number of acute medication intake. General inclusion criteria were age between 18 and 65 years old, headache occurring on ≥15 days/four weeks, patients without any medical condition that might put them at increased risk if exposed to botulinumtoxinA (e.g., myasthenia gravis, Eaton–Lambert syndrome, amyotrophic lateral sclerosis, any other significant disease that could interfere with neuromuscular function), no other primary/secondary headache disorder, Beck’s Depression Inventory score of <24 at day 1 of baseline, no neuro-ophthalmological disorders (verified by the assessment of visual acuity, intraocular pressure measurement, and indirect ophthalmoscopy), and no previous exposure to any botulinum toxin serotype. General exclusion criteria were women who were pregnant or in lactation and prophylactic treatment in the previous 3 months.

All patients enrolled in the study filled headache diaries daily (mailed when enclosed in the waiting list for consulting) for at least one month before attending the first visit and for two months after. The project was approved by the ethical review board of the Faculty of Medicine, Sapienza University of Rome, Italy (RIF.CE: 4102), and all subjects gave written informed consent to participate in the study. The patients received a comprehensible oral and written informed consent prior to any specific study procedures, in accordance with the Declaration of Helsinki and local Ethics Committee.

### 5.2. Procedure

All recording sessions were conducted using a Digitimer D360 amplifier (band-pass 0.05–2000 Hz, Gain 1000) and a CED^TM^ power 1401 analogue-to-digital converter (Cambridge Electronic Design Ltd., Cambridge, UK). All equipment was connected to a computer running Windows 8, on which the CED Signal v11 software was running. The study participants were made to sit comfortably in an armchair, placed in a quiet room and instructed to remain as relaxed as possible, with their eyes open. The examiners (G.S. and F.C.) continuously checked the level of attention and vigilance during the recording sessions, which took place in the afternoon between 2 p.m. and 7 p.m. The recordings were made at time 0 (T0), i.e., just before the injection, and at 1 month (T1) and 3 months (T3) after an infiltrative session with BoNTA (Figure 3) in accordance with the PREEMPT protocol [26]. In short, 31 fixed-site, fixed-dose, intramuscular injections (minimum dose: 155 U) were given for the trial over seven distinct head/neck muscle regions (corrugator, procerus, frontalis, temporalis, occipitalis, cervical paraspinal and trapezius). A follow-the-pain strategy with additional dose (up to 40 U) was permitted per protocol depending on the location(s) of the patient’s primary pain (“follow-the-pain” method) and level of palpable muscle tenderness.

### 5.3. Nociceptive Blink (nBR), Trigemino-Cervical Reflexes (nTCR) and Pain-Related Evoked Potentials (PREP) Recordings

Percutaneous electrical stimulation of the innervation territory of the supraorbital nerve (SON) at the forehead was obtained by means of a nociception-specific concentric surface electrode, constructed according to the physical characteristics described by Kaube et al. (2000) [39] (see Serrao et al., 2010 for technical details [28]). The stimulation consisted of a train of electrical stimuli composed of three pulses, each of 0.1 ms duration (inter-pulse interval 5 ms) [60]. Electromyographic signals were recorded from both orbicularis oculi muscles with electrodes placed infraorbitally (active) and latero-orbitally (reference) [61], and from the semispinalis capitis muscle on the right side (active electrode at the C3 level, reference on the C7 spinous process [28]). During the noxious supraorbital nerve stimulation pain-related cortical evoked potentials (PREP) were simultaneously recorded at Cz (10–20 international system) using as reference linked ear-lobes [62]. The recordings were performed with the subjects comfortably seated on an armchair; head and neck positions minimizing neck muscle activity were chosen.

The EMG and cortical activities were carried out using an analysis time window of 500 ms after the delivery of the electrical stimulus. 

The following parameters were measured both for nBR and nTCRs: sensory thresholds (detection and pain), latency, area under the curve (AUC), and its habituation. Latency, peak-to-peak amplitude, and habituation were also measured for PREP. 

#### 5.3.1. Sensory Thresholds

Individual sensory detection (ST) and pain (PT) thresholds were defined as the minimum stimulation intensity detected as tactile or perceived as painful, respectively, over three series of ascending and descending stimulus intensities.

#### 5.3.2. Recordings

A stimulus intensity fixed at 1.5× PT was used to record the nociceptive reflexes; EMG signals were recorded, averaged, and full wave rectified, after the application of a 10 Hz high-pass digital filter. In all subjects, the first recording sweep was discarded to avoid contamination by an initial startle response. According to previous publications, we used fixed onset and offset for the calculation of the nBR area under the curve (AUC: µV × ms), by positioning the cursors between 27 and 87 ms post-stimulus [39]. For the nTCR, the onset and offset latencies were defined as the time point at which the amplitude of the reflex EMG signal exceeded by more than 30 uV the background EMG activity and returned below this level. The reflex duration and AUC were measured between these two points (onset, offset) [28]. For the PREP, after applying a digital low-pass 100 Hz filter, we measured the latency and the amplitude of the negative-positive vertex complex (N-P).

#### 5.3.3. Habituation

To assess habituation, we recorded three blocks of six responses with an interstimulus interval of 40 s; the interval between blocks was 2 min. The first sweep of each block was excluded from further analysis to avoid startle response contamination. The mean AUC values of the nBR and nTCR, and the N-P PREP amplitude were analyzed off-line for each block of recordings and averaged to calculate habituation, defined as the slope of the linear regression of the AUC (nBR and nTCR) or the amplitude (PREP) between the first and the second or third block of recordings.

### 5.4. Somatosensory Evoked Potentials (SSEPs)

SSEPs were elicited by an electrical stimulus applied to the right median nerve at the wrist using a constant current square wave pulse (0.1 ms width, cathode proximal). The stimulus intensity was set at 1.2 times the motor threshold and the repetition rate at 4.4 Hz. The active recording electrodes were placed over the contralateral parietal area (C3′, 2 cm posterior to C3 in the International 10–20 system), on the fifth cervical spinous process (Cv5), both referenced to Fz, and on the Erb’s point on the stimulated side, referenced to the contralateral Erb’s point; the ground electrode was placed on the right arm [63]. 

In brief, while subjects were asked to keep their eyes open and to fix attention on the stimulus-induced thumb movement, we collected 300 artefact-free sweeps of 50 ms (5000 Hz sampling rate), during 4.4 Hz continuous stimulation. A digital low pass filter at 450 Hz was applied off-line.

We considered as “grand average”, the average of 300 artefact-free evoked responses. The various SSEP components (N9, N13, N20, P25 and N33) were identified according to their respective latencies. We measured peak-to-peak amplitudes of the peripheral N9, the cervical N13 and the cortical N20-P25 and P25-N33 components. 

Thereafter, the 300 evoked responses were partitioned in 3 sequential blocks of 100 responses. Each block was averaged off-line (“block averages”) and analyzed for N20-P25 amplitudes. We calculated habituation as the slope of the linear regression of the N20-P25 SSEP amplitude between the 1st and the 2nd or 3rd block of recordings.

### 5.5. Clinical Data

From the headache diaries, we calculated the percentage changes in monthly days with headache, mean severity of headache (on a visual analogue scale) and monthly number of acute medication intake.

### 5.6. Statistical Analyses

Statistical analyses were performed by an independent operator to whom clinicians and neurophysiologists independently forwarded their data. Based on previous pharmacological study [64], we had planned a sample size of 20 patients, but enrolled only 15 patients because botulinum toxin type A became unavailable in our hospital. All statistical analyses were performed using SPSS for Windows, version 21. Neurophysiological parameters were analyzed with Anderson–Darling’s and/or Kolmogorov–Smirnov’s tests for normal distribution at each time point of recordings. If distribution was normal, we used ANOVA and between-group Tukey’s multiple comparisons method, and if not, we used Friedman’s non-parametric test and between-group Dunn–Bonferroni multiple comparisons method [65]. 

We searched for correlations between electrophysiological and clinical data using Pearson’s test for Gaussian distributed data, otherwise we used Spearman’s tests.

For all inferential statistics, a *p*-value of less than 0.05 was considered significant.

## Figures and Tables

**Figure 1 toxins-15-00076-f001:**
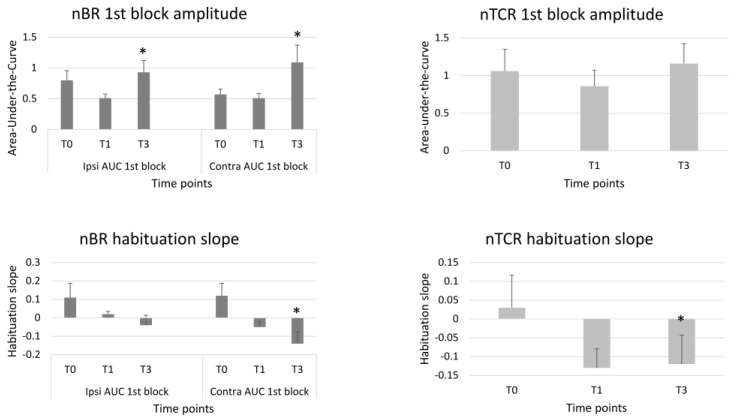
1st block amplitudes and habituation slopes of the ipsi- and contralateral nociceptive blink reflex (nBR) and of the nociceptive trigemino-cervical reflex (nTCR) before (T0), 1 month (T1) and 3 months (T3) after the BoNTA injections. Means ± standard error of means. * = *p* < 0.05 vs. T0.

**Figure 2 toxins-15-00076-f002:**
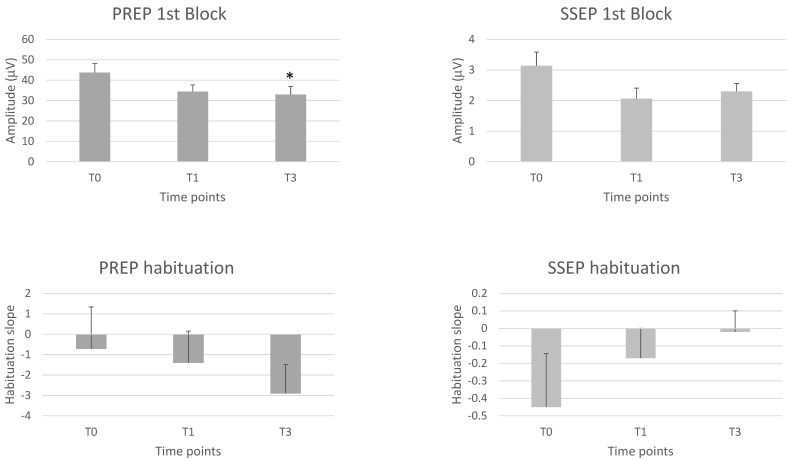
1st block amplitudes and habituation slopes of the trigeminal pain-related cortical evoked potential (PREP) and the upper limb somatosensory evoked cortical potential (SSEP) before (T0), 1 month (T1) and 3 months (T3) after the BoNTA injections. Means ± standard error of means. * = *p* < 0.05 vs. T0.

**Figure 3 toxins-15-00076-f003:**
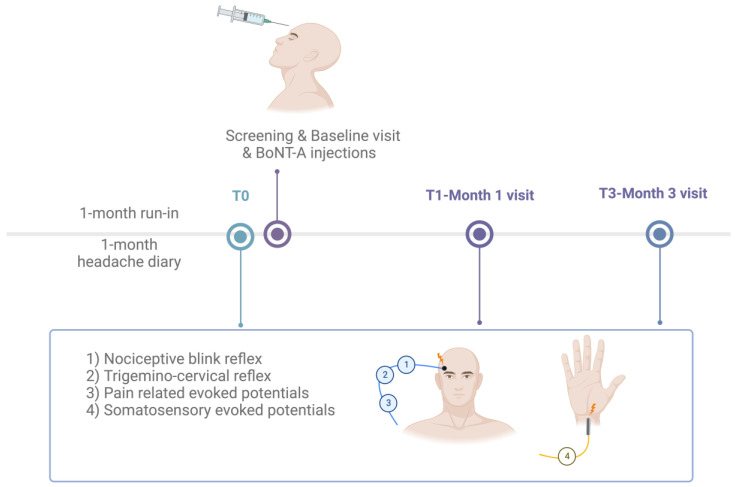
Flow chart of the study’s visits and recording sessions. Chronic migraine patients were recorded just before (T0), 1 month (T1) and 3 months (T3) after one session of BoNT-A injections. The nociceptive trigeminal system was tested through the recording of nociceptive blink reflex (1), trigemino-cervical reflex (2), and pain-related evoked potentials (3), during the percutaneous electrical stimulation of the innervation territory of the supraorbital nerve at the forehead. The non-painful lemniscal sensory systems were investigated through somatosensory evoked potentials (4) elicited by electrically stimulating the right median nerve at the wrist (Created with BioRender.com, accessed on 23 December 2022).

**Table 1 toxins-15-00076-t001:** Descriptive statistics for clinical variables: median and interquartile range (25–75%); inferential statistics based on Friedman’s tests S, *p* values; N = number. Results that are statistically significant are highlighted in bold.

	T0	T1	T3	(S; *p*)
Days with headache/month (N)	30.0 20.0; 2300	14.0 10.0; 15.0	10.0 9.75; 14.25	**24.15; <0.001**
Headache severity (0–3)	8.0 8.0; 10.0	8.0 6.0; 8.0	7.0 6.0; 8.0	**13.65; 0.001**
Tablets/month (N)	50.0 30.0; 78.0	13.0 8.0; 20.0	10.0 5.0; 14.0	**21.28; <0.001**

**Table 2 toxins-15-00076-t002:** Descriptive statistics for nociceptive blink reflex (nBR) parameters: median and interquartile range (25–75%); inferential statistics based on ANOVA; ^§^ symbol denotes the application of Friedman’s tests S, *p* values; AUC = area under the curve. Results that are statistically significant are highlighted in bold.

	T0	T1	T3	(F or S; *p*)
nBR Sensory threshold (mA)	2.0 2.0; 3.45	2.55 2.0; 3.0	2.5 2.0; 3.5	1.02; 0.599 ^§^
nBR Pain threshold (mA)	8.5 6.0; 11.0	9.0 7.0; 12.0	11.0 10.0; 15.0	**3.23; 0.050**
Ipsilateral 1st block AUC	0.578 0.396; 0.701	0.456 0.300; 0.700	0.746 0.416; 1.254	**6.00; 0.050 ^§^**
Ipsilateral 2nd block AUC	0.638 0.498; 1.513	0.496 0.322; 0.714	0.693 0.330; 1.336	4.93; 0.085 ^§^
Ipsilateral 3rd block AUC	0.589 0.351; 1.199	0.420 0.300; 0.700	0.651 0.223; 1.111	3.00; 0.223 ^§^
Ipsilateral habituation slope 2nd block	0.030 −0.013; 0.22	−0.025 −0.02; 0.060	−0.050 −0.09; 0.009	1.98; 0.151
Ipsilateral habituation slope 3rd block	−0.005 −0.085; 0.11	−0.005 −0.04; 0.025	−0.035 −0.05; 0.035	1.28; 0.528 ^§^
Contralateral 1st block AUC	0.443 0.335; 0.584	0.471 0.351; 0.558	0.721 0.337; 1.358	**5.31; 0.010**
Contralateral 2nd block AUC	0.422 0.362; 0.688	0.371 0.326; 0.640	0.606 0.295; 1.049	**3.51; 0.040**
Contralateral 3rd block AUC	0.377 0.226; 0.501	0.375 0.228; 0.511	0.651 0.283; 1.073	**3.46; 0.042**
Contralateral habituation slope 2nd block	0.070 −0.080; 0.180	−0.030 −0.130; 0.030	−0.080 −0.320; 0.030	**6.03; 0.049 ^§^**
Contralateral habituation slope 3rd block	−0.010 −0.050; 0.070	-0.063 −0.084; −0.139	−0.035 −0.141; −0.010	1.19; 0.316

**Table 3 toxins-15-00076-t003:** Descriptive statistics for nociceptive trigeminocervical reflex (nTCR) parameters: median and interquartile range (25–75%); inferential statistics based on ANOVA; ^§^ symbol denotes the application of Friedman’s tests S, *p* values; AUC = area under the curve. Results that are statistically significant are highlighted in bold.

	T0	T1	T3	Statistics
nTCR onset (ms)	71.05 58.73; 78.23	68.99 52.74; 76.50	71.05 62.94; 80.15	0.41; 0.664
nTCR duration (ms)	102.59 87.85; 123.23	93.75 86.66; 102.18	91.98 83.22; 100.83	1.23; 0.303
nTCR grand-average AUC	0.378 0.298; 0.968	0.272 0.217; 0.67	0.280 0.235; 0.993	2.60; 0.273 ^§^
nTCR 1st block AUC	0.512 0.291; 1.368	0.406 0.269; 1.308	1.070 0.224; 1.401	2.22; 0.330 ^§^
nTCR 2nd block AUC	0.475 0.335; 1.888	0.307 0.214; 0.925	0.340 0.193; 1.096	2.00; 0.368 ^§^
nTCR 3rd block AUC	0.310 0.237; 0.819	0.218 0.185; 0.616	0.249 0.190; 0.888	0.50; 0.779
nTCR habituation slope 2nd block	0.020 −0.100; 0.400	−0.040 −0.105; 0.015	−0.050 −0.170; 0.035	1.38; 0.500 ^§^
nTCR habituation slope 3rd block	0.005 −0.145; 0.085	−0.060 −0.128; 0.015	−0.125 −0.205; 0.005	**5.71; 0.028 ^§^**

**Table 4 toxins-15-00076-t004:** Descriptive statistics for pain-related evoked potentials (PREPs) parameters: median and interquartile range (25–75%); inferential statistics based on ANOVA; ^§^ symbol denotes the application of Friedman’s tests S, *p* values; ms = milliseconds. Results that are statistically significant are highlighted in bold.

	T0	T1	T3	Statistics
N latency (ms)	130.0 120.00; 134.00	128.54 123.82; 133.00	128.27 124.71; 132.37	2.85; 0.240 ^§^
P latency (ms)	215.00 179.83; 239.39	212.85 180.42; 227.59	212.85 179.83; 221.00	0.22; 0.802
N-P 1st amplitude block (μV)	38.64 29.39; 57.81	31.88 21.61; 46.64	29.80 21.19; 39.35	**3.40; 0.043**
N-P 2nd amplitude block (μV)	43.89 32.86; 59.76	31.99 25.00; 43.89	31.00 20.79; 34.47	**4.77; 0.014**
N-P 3rd amplitude block (μV)	39.16 19.08; 57.86	30.44 20.24; 45.95	25.38 19.30; 30.31	3.18; 0.053
PREP habituation slope 2nd block	2.91 −0.103; 12.48	−1.02 −0.823; 7.42	−0.56 −5.02; 3.70	2.87; 0.069
PREP habituation slope 3rd block	0.03 −3.58; 5.64	−1.39 −2.71; 3.98	−2.19 −5.06; 0.16	0.30; 0.746

**Table 5 toxins-15-00076-t005:** Descriptive statistics for somatosensory evoked potentials (SSEPs) parameters: median and interquartile range (25–75%); inferential statistics based on ANOVA. ^§^ symbol denotes the application of Friedman’s tests S, *p* values; ms = milliseconds.

	T0	T1	T3	(F, *p*)
SSEP MT (mA)	9.0 8.0; 11.0	9.0 8.0; 10.5	9.0 8.0; 10.5	3.59; 0.166 ^§^
N9 latency (ms)	9.93 9.40; 10.19	9.81 9.40; 10.17	9.98 9.32; 10.64	0.06; 0.937
N13 latency (ms)	13.45 12.81; 13.58	13.15 12.86; 13.57	13.15 12.97; 13.41	0.04; 0.959
N20 latency (ms)	19.17 18.40; 19.80	18.98 18.50; 19.23	18.93 18.48; 19.39	0.00; 0.997
P25 latency (ms)	24.17 22.55; 26.05	23.41 22.03; 26.05	24.53 22.99; 25.89	0.65; 0.525
N33 latency (ms)	32.00 30.67; 34.40	32.47 30.58; 30.80	32.44 30.58; 33.00	0.01; 0.989
N9 amplitude (μV)	2.81 1.44; 3.42	2.64 2.00; 3.37	2.43 2.00; 3.09	0.02; 0.984
N13 amplitude (μV)	1.67 1.47; 1.83	1.85 1.50; 2.08	2.00 1.77; 2.52	4.35; 0.113 ^§^
N20-P25 amplitude (μV)	1.76 1.45; 2.65	1.57 0.97; 2.35	1.57 1.05; 2.39	0.08; 0.921
P25-N33 amplitude (μV)	1.10 0.99; 1.99	1.00 0.72; 1.94	1.22 0.80; 2.22	5.10; 0.078 ^§^
N20-P25 1st block amplitude (μV)	2.79 2.08; 3.87	2.45 1.22; 3.70	2.30 1.46; 3.20	3.73; 0.155 ^§^
N20-P25 2nd block amplitude (μV)	2.59 1.98; 2.98	2.08 1.43; 3.01	2.30 1.40; 3.30	0.51; 0.606
N20-P25 3rd block amplitude (μV)	2.00 1.53; 2.69	1.93 1.65; 2.98	2.25 1.51; 3.00	0.15; 0.926 ^§^
Habituation slope 1–2 block	−0.11 −0.708; 0.388	0.10 −0.35; 0.25	−0.11 −0.50; 0.20	0.45; 0.644
Habituation slope 1–3 block	−0.08 −0.455; 0.035	-0.04 −0.26; 0.21	0.10 −0.25; 0.22	1.28; 0.289

## Data Availability

The data that support the findings of this study are available from the corresponding author upon reasonable request.

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
