# Peer review of "Effects of Botulinum Toxin Type A on the Nociceptive and Lemniscal Somatosensory Systems in Chronic Migraine: An Electrophysiological Study"

_toxins, 2023, doi:10.3390/toxins15010076_

Round 1
Reviewer 1 Report
The paper reports a nicely conducted study assessing the clinical and electrophysiological effects of a single BoNT-A treatment in medication overuse headache and chronic migraine patients. The results are well presented and discussed. The study’s limitations are well acknowledged. I have a few modest and advisory comments.
The details of BoNT-A single treatment are not reported. Although it is said that the single BoNT-A injection was performed according to procedures described elsewhere (Diener et al. 2010, Cephalalgia, 30:804-814), this larger-scale study differs from the present one in many respects. Thus, it would be helpful if a brief description of the BoNT-A treatment were offered.
In the Results section, reference is made to a post-hoc test when reporting statistical results. The suggestion is made to indicate the specific test used.
It is unclear why the same or similar results are presented simultaneously in tabular and graph forms (for instance, Table 1 and Figure 1 both report the same clinical data, although using alternative central and dispersion parameters). This seems to be an unnecessary repetition and might confuse.
In Table 2, T0 and T3 data for ipsilateral and contralateral habituation slope 2nd block are precisely the same. Please, confirm this is not a mistake when writing up the table.
The report of statistical analyses is sometimes confusing. In the Table 1 header, reference is made to ANOVA and Tukey’s tests, but the figures in the table are from Friedman’s test. I understand that Friedman’s test is a non-parametric alternative to repeated measures ANOVA but not an ANOVA test. Also, reference is made to Chi-square statistics (e.g., Table 4), but the use of this test is not mentioned in the Methods section.
The term “repetition” is used to name repeated time measurements. It took me a second reading to make sense of this term. The authors could choose a less ambiguous term to call this factor.
p.9, ln 211. The abbreviation “HV” was not explained in the text before. The same applies to the abbreviation “PT” on p. 10, ln 224.
Author Response
Response to Reviewer 1 Comments
Point 1:
The details of BoNT-A single treatment are not reported. Although it is said that the single BoNT-A injection was performed according to procedures described elsewhere (Diener et al. 2010, Cephalalgia, 30:804-814), this larger-scale study differs from the present one in many respects. Thus, it would be helpful if a brief description of the BoNT-A treatment were offered.
Response 1: We thank the Reviewer for his/her very pertinent comment. Indeed, we added a careful and detailed description about the protocol used for BoNT-A injections in the Methods sections. Now, we have expanded the description about the injection’s protocol as follows (p. 12, lines 359-364): “In short, 31 fixed-site, fixed-dose, intramuscular injections (minimum dose: 155 U) were given for the trial over seven distinct head/neck muscle regions (corrugator, procerus, frontalis, temporalis, occipitalis, cervical paraspinal and trapezius). A follow-the-pain strategy with additional dose (up to 40 U) was permitted per protocol depending on the location(s) of the patient's primary pain ("follow-the-pain" method) and level of palpable muscle tenderness”
Point 2:
In the Results section, reference is made to a post-hoc test when reporting statistical results. The suggestion is made to indicate the specific test used.
Response 2: In order to simplify the readability of the manuscript, in the Results section, we have reworded the “post-hoc test” with “between group” to show statistical inferences between pairs of groups.
As now specified in paragraph 5.6 “Statistical analyses”, post hoc tests were Tukey and Dunn-Bonferroni, respectively for ANOVA and Friedman. The Statistical Analyses paragraph was integrated as follows (p. 15, lines 451-452): “……Friedman’s non-parametric test and between-group Dunn-Bonferroni multiple comparisons method [65].”, and we added the between group test in the Table 2 and 3.
We have added one more reference to the list:
- Dunn OJ (1964) Multiple Comparisons Using Rank Sums. Technometrics 6:241–252. https://doi.org/10.1080/00401706.1964.10490181
Point 3:
It is unclear why the same or similar results are presented simultaneously in tabular and graph forms (for instance, Table 1 and Figure 1 both report the same clinical data, although using alternative central and dispersion parameters). This seems to be an unnecessary repetition and might confuse.
Response 3: Following the Reviewer suggestion, now we have removed Figure 1 and 2. We prefer to leave Figure 3 and 4 because may be helpful to graphically highlights the significant study’s results.
Point 4:
In Table 2, T0 and T3 data for ipsilateral and contralateral habituation slope 2nd block are precisely the same. Please, confirm this is not a mistake when writing up the table.
Response 4: The Reviewer is right, we checked and discovered that we made a mistake during the copy/paste from the statistical analysis software. We added correct data in Table 2. The statistics was already correct, then not modified.
Point 5:
The report of statistical analyses is sometimes confusing. In the Table 1 header, reference is made to ANOVA and Tukey’s tests, but the figures in the table are from Friedman’s test. I understand that Friedman’s test is a non-parametric alternative to repeated measures ANOVA but not an ANOVA test. Also, reference is made to Chi-square statistics (e.g., Table 4), but the use of this test is not mentioned in the Methods section.
Response 5: The Reviewer is right. Since in Table 1 all tests are non-parametric, now we replaced “ANOVA and Tukey’s tests” in the header with “Friedman’s test S” (p. 2, line 83). The post-hoc test of Freadman is the Dunn-Bonferroni test actually is based on Chi-square statistics (see the new reference 65). Hovewer, to avoid misunderstandings we replaced “Chi-square” with “Dunn-Bonferroni” (p. 3, line 99).
Point 6:
The term “repetition” is used to name repeated time measurements. It took me a second reading to make sense of this term. The authors could choose a less ambiguous term to call this factor.
p.9, ln 211. The abbreviation “HV” was not explained in the text before. The same applies to the abbreviation “PT” on p. 10, ln 224.
Response 6: To avoid confusion, now we have replaced the term “repetition” with “repeated time measurements” all along the Results section.
We also replaced the abbreviation “HV” with “healthy volunteers”, while the abbreviation PT on p.10 was already explained before in the manuscript (p.10, line 234).
Reviewer 2 Report
The manuscript “Effects of botulinum toxin type A on the nociceptive and lemniscal somatosensory systems in chronic migraine: an electrophysiological study” by XYZ et al. They have reported the effects of botulinum toxin type A against chronic migraine using an electrophysiological study. The authors have done various parameters to prove their hypothesis. The manuscript is written well. After thoroughly reviewing I feel the manuscript needs to be revised.
Comments:
1. In the abstract section, I will suggest rewriting the conclusion in a better way.
2. I will suggest citing the latest literature in the introduction section and rewriting it properly and adding a few more sentences about botulinum toxin type A.
3. I will suggest explaining properly the discussion section.
4. To show the statistical significance authors have used (*), I will suggest authors explain the (*) meanings and how it is related to different groups.
5. How was the dose of BoNT-A was decided?
6. I will suggest adding a graphical abstract as it can be a point of attraction to the readers.
Author Response
Response to Reviewer 2 Comments
Point 1:
In the abstract section, I will suggest rewriting the conclusion in a better way.
Response 1: In order to clarify and make it easier to understand, we have revised the conclusion in the abstract section as follows: “Our study provides electrophysiological evidence for the ability of a single session of BoNT-A injections to exert a neuromodulatory effect at the level of trigeminal system through a reduction of input from meningeal and other trigeminovascular nociceptors. Moreover, by reducing activity in cortical pain processing areas, BoNT-A restores normal functioning of the descending pain modulation systems.” (p. 1, lines 17-22).
Point 2:
I will suggest citing the latest literature in the introduction section and rewriting it properly and adding a few more sentences about botulinum toxin type A.
Response 2: According to the reviewer suggestion, in the introduction section we added some citation of recent articles (between 2020 and 2022) about the usage of BoNT-A in the treatment of chronic migraine.
Citations:
- Becker WJ (2020) Botulinum Toxin in the Treatment of Headache. Toxins (Basel) 12:. https://doi.org/10.3390/toxins12120803.
- Kępczyńska K, Domitrz I (2022) Botulinum Toxin-A Current Place in the Treatment of Chronic Migraine and OtherPrimary Headaches. Toxins (Basel) 14: https://doi.org/10.3390/toxins14090619.
Moreover, we added few more sentences about botulinum toxin type A in the Introduction as follows (p. 1, lines 38-42): “OnabotulinumtoxinA (BoNT-A) has been shown to alleviate pain in a number of conditions, including migraine [12]. The PREEMPT (Phase III REsearch Evaluating Migraine Prophylaxis Therapy) clinical trial evaluated the safety and efficacy of BoNT-A in adult migraine patients and found that, compared with placebo, it reduced the mean frequency of headache days [13].”
Point 3:
I will suggest explaining properly the discussion section.
Response 3: Now we have double checked the Discussion section, simplifying, and revising the English style.
Point 4:
To show the statistical significance authors have used (*), I will suggest authors explain the (*) meanings and how it is related to different groups.
Response 4: We aggreed with the suggestion made by the Reviewer and we modified the (*) in the tables where was substituted by (§) symbol which denotes the application of Friedman’s tests S (explained in the tables caption), while the (*) remained in figures where it highlight statistical significant results (explained in the figure captions with “* = p < 0.05”). Furthermore, we have made it clearer in the figure legend that the asterisk refers to the comparison to the value at T0. Added in the caption as follows: "* = p < 0.05 vs T0".
Point 5:
How was the dose of BoNT-A was decided?
Response 5: We appreciate the Reviewer for his very pertinent comment. Indeed, we added a careful and detailed description about the protocol used for BoNT-A injections (PREEMP-Protocol) in the Methods sections. (p. 12, lines 359-364) “In short, 31 fixed-site, fixed-dose, intramuscular injections (minimum dose: 155 U) were given for the trial over seven distinct head/neck muscle regions (corrugator, procerus, frontalis, temporalis, occipitalis, cervical paraspinal and trapezius). A follow-the-pain strategy with additional dose (up to 40 U) was permitted per protocol depending on the location(s) of the patient's primary pain ("follow-the-pain" method) and level of palpable muscle tenderness.”
Point 6:
I will suggest adding a graphical abstract as it can be a point of attraction to the readers.
Response 6: We appreciated and agreed with this suggestion. Now, we have added the required graphical abstract to the manuscript.
Reviewer 3 Report
The manuscript entitled “Effects of botulinum toxin type A on the nociceptive and lemniscal somatosensory systems in chronic migraine: an electrophysiological study” is well written and clear for the reader. The authors carefully described the experimental design and properly commented results. The manuscript is worthy of publication after minor revisions. The authors should improve the table and figures captions detailing their contents. In Figure 4, the authors mention Chi-square. The authors did not mention this test in the statistical analysis section. Please check for this in the manuscript and change the text/table if proper. The authors used msec in the table and along the text to express milliseconds. Please check tables and text and change msec with ms.
Author Response
Response to Reviewer 3 Comments
Point 1:
The authors should improve the table and figures captions detailing their contents. In Figure 4, the authors mention Chi-square. The authors did not mention this test in the statistical analysis section. Please check for this in the manuscript and change the text/table if proper.
Response 1: The Freadman’s post hoc-test is actually a chi-square-based test (Dunn-Bonferroni, see the added reference 65). However, as replyed to Reviewer #1, to avoid misunderstandings we replaced “Chi-square” with “Dunn-Bonferroni” (p. 3, line 99). As suggested by the Reviewer, we improved table and figures captions, especially that of Figure 3 (as requested by the Reviewer #1, we have removed redundant Figures 1 and 2).
Point 2:
The authors used msec in the table and along the text to express milliseconds. Please check tables and text and change msec with ms.
Response 2: We thank the Reviewer for the pertinent observation, therefore we have modified the text and tables as suggested.